# Association of Gut Microbiota Composition in Pregnant Women Colonized with Group B *Streptococcus* with Maternal Blood Routine and Neonatal Blood-Gas Analysis

**DOI:** 10.3390/pathogens11111297

**Published:** 2022-11-04

**Authors:** Zhixia Wang, Wenyuan Pu, Qi Liu, Meifeng Zhu, Qinlei Chen, Yingchun Xu, Chunxiang Zhou

**Affiliations:** 1School of Chinese Medicine, School of Integrated Chinese and Western Medicine, Nanjing University of Chinese Medicine, Nanjing 210024, China; 2Department of Gynecology and Obstetrics, BenQ Medical Center, The Affiliated BenQ Hospital of Nanjing Medical University, Nanjing 210019, China; 3Department of Nephrology, Changzhou Affiliated Hospital of Nanjing University of Chinese Medicine, Changzhou 213000, China; 4Department of Pediatrics, BenQ Medical Center, The Affiliated BenQ Hospital of Nanjing Medical University, Nanjing 210019, China; 5Department of Chinese Medicine, BenQ Medical Center, The Affiliated BenQ Hospital of Nanjing Medical University, Nanjing 210019, China

**Keywords:** Group B *Streptococcus*, microbiome, blood cell count, umbilical arterial blood-gas analysis

## Abstract

Group B *Streptococcus* (GBS) colonizes the vaginal and rectal mucosa in a substantial proportion of healthy women, and GBS is a risk factor for GBS-associated adverse birth outcomes, such as bacterial infection, in neonates. Whether changes in the gut microbiota of GBS-infected pregnant women are associated with maternal complete blood cell count (CBC) and neonatal blood-gas analysis is unknown. To explore the relationship between the intestinal microecological composition of pregnant women and maternal blood routine and neonatal blood-gas analysis, we collected intestinal microecology samples of 26 pregnant women in clinic. They were divided into a positive group(GBS positive,GBS +) and a negative group (GBS negative, GBS-), with 12 in the positive group and 14 in the negative group. 16S rRNA gene sequencing was used to examine the gut microbiota profile from a fecal sample of pregnant women. CBC was carried out in enrolled pregnant women and umbilical arterial blood-gas analysis (UABGA)was conducted for analysis of intestinal microbiota composition, maternal blood routine and neonatal blood gas. Our results showed significant differences in the total number of organisms and microbial diversity of intestinal microbiota between healthy pregnant women and GBS-positive pregnant women. Particularly, abundances of *Lentisphaerae, Chlorobi, Parcubacteria, Chloroflexi, Gemmatimonadetes, Acidobacteria, Fusobacteria* and *Fibrobacteres* were only detected in participants with GBS colonization. Blood-gas analysis revealed that neonates born to mothers with GBS colonization had significantly higher fractions of carboxyhemoglobin (FCOHb) and lower methemoglobin (FMetHb), and abundances of OTU80, OTU122, OTU518 and OTU375 were associated with blood-gas indicators, such as carboxyhemoglobin, methemoglobin, PCO2, PH and ABE. Interestingly, there were significant correlations between OTU levels and inflammatory indexes in pregnant women with GBS infection. Together, this study revealed for the first time that altered gut microbiota compositions are related to the inflammatory state in GBS-positive pregnant women and neonatal blood-gas indicators. GBS colonization may lead to significant changes in the gut microbiome, which might be involved in the pathogenesis of the maternal inflammatory state and neonatal blood gas abnormalities.

## 1. Introduction

Group B *Streptococcus* (GBS), also known as *Streptococcus agalactiae*, is a facultative anaerobic gram-positive coccus that can intermittently, transiently or persistently colonize the digestive tract and reproductive tract [1]. Pregnant women colonized with GBS can transmit it to the fetus or the newborn during or after delivery, causing life-threatening infections in newborns, such as pneumonia, meningitis, sepsis or/and death [2]. To date, GBS infection is still one of the most common causes of vertically transmitted perinatal bacterial infection around the world [2]. Early-onset GBS infection in newborns occurs within 7 days after birth, accounting for 80% of neonatal GBS infections [3]. Neonatal early-onset GBS infection is clinically characterized by pneumonia, sepsis and meningitis; neonates with pneumonia usually develop it within 12 to 24 hours after birth, and the clinical manifestations are similar to those of neonatal respiratory distress syndrome [3]. Neonatal late-onset GBS infection often occurs 7 days of birth with nosocomial cross-infection and horizontal transmission, and this type is mainly manifested as meningitis, bacteremia and sepsis [3]. Neonatal GBS infection may suggest a poor prognosis when coma, convulsions, hypotension and apnea are present, and approximately 15% to 30% of surviving infants may suffer from serious sequelae, mainly including hydrocephalus, movement disorder, intellectual disability, ventriculitis, hemiplegia or general paralysis, epilepsy, language disorder, cortical blindness and deafness [4]. Therefore, clarifying the distribution characteristics and underlying mechanism of GBS colonization will help us to develop preventive and therapeutic approaches for GBS infection.

In recent years, many studies have shown that imbalance of intestinal microorganisms is related to chronic inflammation, development, carcinogenesis and immune evasion [5]. Coincidentally, the gut microbiota has been suggested to be involved in the potential intrauterine and early development of infants [6,7]. It is well known that early patterns of gut colonization can predispose children to disease risk later in life; thus, the involvement of the gut microbiome in neonatal diseases may be an area of interest [8]. The evidence for the relationship between maternal GBS infection and the infant gut microbiota is emerging. For instance, the relative abundances of *Clostridiaceae*, *Ruminococcoceae* and *Enterococcaceae* tend to be higher in infants of GBS-positive mothers compared to those of GBS-negative ones [9]. Moreover, pregnant women carrying GBS have differences in the vaginal microbiome when compared GBS-negative women [10,11]. It is highly probable that the presence of GBS alters the vaginal microbiome of pregnant women, and this could also affect the early microbial exposures encountered at birth, the development of the infant’s gut microbiome and the risk of future disease in offspring [12]. Here, one should bear in mind that analyses focusing only on the larger microbiota state types may limit the assessment of known pregnancy pathogenic microbial species that are less abundant. Improved understanding of maternal GBS colonization could provide novel insights into the prevention of GBS complications with maternal and fetal health. While there are growing studies on the gut microbiota composition in fecal samples of GBS-positive neonates, there remains the need to determine whether changes in the gut microbiome of pregnant women with GBS colonization might have an impact on the prenatal or early-life determinants of the infants. To this end, we used 16S rRNA sequencing to detect the compositions of fecal microbiota in GBS-positive and GBS-negative pregnant women, and investigated the association of the abnormal microbiota with neonatal birth outcomes and blood-gas analysis. 

## 2. Materials and Methods

### 2.1. Enrollment of Patients and Healthy Controls

In this study, we collected a total of 26 pregnant women’s stool samples due to difficult clinical sampling, and grouped them according to the results of their GBS screening, including 12 in the GBS-positive group and 14 in the GBS-negative group (healthy controls). All participants were authorized by the ethics committee and signed the informed consent form.

GBS colonization status was tested for pregnant women from 35 to 37 weeks of gestation in the outpatient maternity examination. Sampling method: take a sterile swab into the vagina without using the vaginal speculum, and then sample within the rectum through the rectal sphincter by another sterile swab. Screening was performed using the DL-GBS B cluster strecoccal culture analysis system(DL-GBS-96, Zhuhai DL Biotech. Co., Ltd.Zhuhai, Guangdong,China). Medium mainly contained protein, yeast powder, inhibitors and chromogenic substances, which is Group B *streptococcus*’s selective color rendering medium. It can provide higher Group B *streptococcus* growth and reproduction of nutrients and growth factors and, at the same time inhibit the growth of miscellaneous bacteria. Group B *streptococcus* is rapidly grown in the medium and color to achieve screening and identification effect. By placing the inoculated medium into the detection system and continuously culturing at a constant temperature of 37 ℃ ± 1.5 ℃, the medium provides high nutrient components and growth factors for growth and reproduction, while inhibiting the growth of some miscellaneous bacteria. Group B *streptococcus* changed the bottom of the medium during the culture. The culture analysis system used optical sensors to continuously monitor the color change at the bottom of the culture medium, so as to determine the negative and positive results of the specimens to be tested, and the color-positive subjects was judged as GBS-colonized.

### 2.2. Sample Collection of Feces

Intestinal microbiota samples were collected in our outpatient clinic. They were usually performed on the day of prenatal examination.Sampling method: collect immediately after stool in morning, keep in a sterile test tube, and then move to a refrigerator of minus 80 degrees Celsius; prepare for testing.

### 2.3. Measurement of CBC in Pregnant Women

All pregnant women are routinely tested for CBC at 37 to 38 weeks to identify anemia and other conditions before labor and birth. The samples were sent to a laboratory within 10 min of collection. CBC test was checked in our hospital clinical laboratory with Sysmex XN-2000 (Sysmex Corporation, Kobe,Japan).

### 2.4. Neonatal Umbilical Artery Blood-Gas Analysis

After the delivery of the newborn, the two ends of the umbilical cord of the blood section were clamped with hemostatic forceps and kept refrigerated at 4 degrees Celsius. PH, oxygen pressure, carbon dioxide pressure, BE and lactic acid values were measured within 20 minutes after birth by Radiometer ABL90FLEX (Radiometer Medical ApS, Bronshoj, Denmark).

### 2.5. 16S rRNA Gene Sequence Analysis

Microbial community genomic DNA was extracted from the samples using the E.Z.N.A.® soil DNA Kit (Omega Bio-tek, Norcross, GA, USA). The DNA purity and concentration tests were performed using the NanoDrop2000 (Thermo Scientific, Wilmington, NC, USA). DNA integrity was determined by 1% agarose gel electrophoresis with a voltage of 5 V/cm for 20 min. The PCR amplification of 16S rRNA genes was performed. Sequencing was conducted with an Illumina Miseq PE300 (Majorbio Bio-Pharm Technology Co.Ltd. Shanghai,China).The raw 16S rRNA gene-sequencing reads were demultiplexed, quality-filtered and merged by Fast Length Adjustment of SHort reads (FLASH) software. Operational taxonomic units (OTUs) with 97% similarity cutoff were clustered using UPARSE version 7.1 [13],and chimeric sequences were identified and removed. The taxonomy of each OTU representative sequence was analyzed by RDP Classifier version 2.2 [14] against the 16S rRNA database (e.g., Silva v138) using a confidence threshold of 0.7. Alpha and beta diversity were generated with the Quantitative Insights Into Microbial Ecology (QIIME) [15] pipeline and computed. The linear discriminant analysis (LDA) effect size (LEfSe) method was used to identify species with statistically significant differential abundance among groups (Segata et al., 2011). The Venn diagram visually shows the similarities and overlap of species composition in different environmental samples, which was used to count the number of total and unique species in multiple groups or multiple samples. In addition, we used partial-least squares discrimination analysis (PLS-DA) and non-metric multi-dimensional scaling (NMDS) [15] to analyze bacterial beta diversity. The relative abundance of phyla, classes, orders, families and genera in all samples was calculated and compared between each groups.

### 2.6. Statistical Analysis 

The results in this study were defined as the mean ± SD. The Kolmogorov–Smirnov test was used to evaluate alpha diversity. Independent t-tests were used to compare the number of gut microbiome samples between GBS-positive and GBS-negative groups. The differences and commonalities among 16S rRNA genes were compared. An analysis of similarities (ANOSIM) on beta diversity matrices was conducted in QIIME. We used the test to check significant differences between the microbial communities. Statistical dependence between continuous variables was examined by Spearman’s rank correlation. 

## 3. Results

As shown in Table 1, the results of demographic data and blood routine examination showed that there were no significant differences between GBS- and GBS+ participants. 16S rRNA analysis was performed to compare the level of bacterial genus between two groups. The Venn diagram of the GBS- and GBS+ participants is shown in Figure 1A. Our results showed that *Lentisphaerae*, *Chlorobi*, *Parcubacteria*, *Chloroflexi*, *Gemmatimonadetes*, *Acidobacteria*, *Fusobacteria* and *Fibrobacteres* were only detected in GBS+ participants. *Actinobacteria*, *Tenericutes*, *Firmicutes*, *Cyanobacteria*, unclassified_k__norank, *Bacteroidetes*, *Verrucomicrobia*, *Saccharibacteria*, *Synergistetes* and *Proteobacteria* were the dominant phyla in the microbiome. This indicates that the microbial composition of the samples of the two groups of subjects was similar. PLS-DA analysis was carried out to evaluate bacterial beta diversity. GBS- group and GBS+ group samples were separated along the horizontal axis. The OTUs were clustered depending on the presence or absence of GBS infection (Figure 1B). As shown in Figure 1C, community bar plot analysis on genus level shows the relative abundance of bacterial community of GBS- and GBS+ participants. We found multiple differentially abundant bacteria in two groups. *Dorea*, *Holdemanella* and *Oscillibacter* were especially abundant in GBS+ participants (Figure 1D). Bioinformatics analyses showed that individuals with GBS- and GBS+ have distinct microbiota compositions, and identified specific OTUs associated with GBS infection. The four OTUs with top relative abundances were OTU80, OTU122, OTU518 and OTU375. The richness of these OTUs was significantly lower in GBS+ patients (Figure 2A–D). 

To investigate whether maternal GBS infection has an effect on newborns, umbilical artery blood gases were analyzed in neonates delivered by two groups of mothers. Neonates born to mothers with GBS infection had significantly higher fractions of carboxyhemoglobin (FCOHb) and lower methemoglobin (FMetHb) than those born to mothers without GBS infection (Figure 3A,B). There were significant correlations between OTU levels and umbilical artery blood gases. The level of OTU80 was positively correlated with fractions of carboxyhemoglobin (r = 0.459, *p* = 0.021), while a negative correlation was observed for methemoglobin (r = −0.51, *p* = 0.009) (Figure 3C,D). The level of OTU518 was negatively correlated with PCO_2_ (r = −0.4438, *p* = 0.0263) (Figure 3E). The level of OTU122 was positively correlated with PH (r = 0.4133, *p* = 0.04) (Figure 3F). The level of OTU375 was negatively correlated with ABE (r = −0.5089, *p* = 0.0094) (Figure 3G).

Although no statistically significant difference was observed with respect to the routine blood test between GBS+ and GBS- participants (Table 1), there were significant correlations between OTU levels and inflammatory indexes. However, OTU80 was only correlated with neonatal blood gases, and had no significant correlation with maternal inflammatory indicators. The level of OTU122 was positive correlated with monocyte count values (r = 0.3966, *p* = 0.0449) (Figure 4A). The level of OTU518 was positively correlated with basophil count value (r = 0.6096, *p* = 0.0009), while a negative correlation was observed for proportion of monocytes (r = −0.6144, *p* = 0.0008) (Figure 4B,C). The level of OTU375 was negative correlated with hemoglobin level (r = −4169, *p* = 0.0341) (Figure 4D).

## 4. Discussion

This study was based on a retrospective clinical study. Data from a total of 2343 pregnant women who were routinely screened for GBS in the authors hospital from June 2017 to December 2018 were collected and GBS colonization and pregnancy-related outcomes were analyzed. A total of 110 GBS-positive patients were screened, and the GBS colonization rate was 4.69%.

Our results showed that there was a unique gut microbiome profile in the GBS-positive women compared with the GBS-negative ones. In addition, changes in the gut microbiomes were closely associated with inflammatory markers of pregnant women with GBS colonization, and with neonatal blood-gas analysis indicators. Our current study provides a possible clue that GBS colonization-affected gut microbiome malfunction might be a novel driver for the development of neonatal infectious morbidity and mortality, which merits further studies. Studies have consistently shown that maternal GBS colonization is a major risk factor for GBS-related diseases in newborns [16]. Despite the clinical application of intrapartum antibiotic prophylaxis, prevention strategies have not been conducted worldwide, and the clinical incidence of neonatal GBS diseases remains high [17]. Gut microbiota may be an important player in maintaining human health, and dysregulation might be involved in numerous human disorders, including GBS infection [18]. Increasing evidence suggests that the microbiome is an important determinant of vaginal pathogen colonization [19]. Specifically, a significant variation across the communities of the vaginal microbiome was detected in GBS-positive and GBS-negative pregnant women from a cohort study, indicating that the interaction between host and microbiome underpins GBS vaginal colonization [18]. Despite the fact that the composition and diversity of meconium microbiota in newborns from pregnant women with GBS colonization are similar to those of healthy controls, the relative abundance of Lacticasibacillus paracasei is significantly reduced in newborn-derived meconium microbiota in the presence of GBS infection, indicating a role for GBS colonization in the composition of meconium microbiota [20]. However, whether GBS colonization affects the composition of the fecal microbiome in pregnant women and neonatal blood-gas analysis results is relatively less well-known. Therefore, this study aimed to determine the effect of GBS colonization on the fecal microbiome of Chinese women and to examine the correlation of those altered gut microbiota with the blood routine in GBS-positive women and blood-gas parameters in neonates. 

Recently, sequencing 16S rRNA genes from maternal feces and neonatal meconium to examine neonatal microbiota dysbiosis and maternal vaginal microbiota dysbiosis has gained tremendous attention, since both microbiota dysbioses are intimately associated with GBS colonization-related neonatal diseases [21,22]. However, it remains unclear whether the changes in the gut microbiota of GBS-positive pregnant women are correlated with biochemical analysis and fetal blood-gas analysis. From the results of partial least-squares discriminant analysis (PLS-DA), we found that the abundance of *Lentisphaerae*, *Chlorobi*, *Parcubacteria*, *Chloroflexi*, *Gemmatimonadetes*, *Acidobacteria*, *Fusobacteria* and *Fibrobacteres* were only detected in pregnant women with GBS colonization. These results indicate that GBS colonization might affect gut health through directly regulating the composition of the gut microbiome. Previously, the abundance of *Lentisphaerae* was found to be higher in the gut microbiota of those with gestational diabetes mellitus [23]. *Parcubacteria* may be responsible for the development of celiac disease in children, since the abundance of *Parcubacteria* was strongly correlated to serum tumor necrosis factor alpha (TNF-α) in children with celiac disease [24]. Compared to healthy subjects, at the phylum level, the relative abundance of *Chloroflexi* was increased in patients with allergy rhinitis [25]. It was reported that abnormal abundance of *Gemmatimonadetes* may be involved in radiation resistance to pelvic intensity-modulated radiation therapy in patients suffering from cervical cancer [26]. The fecal abundance of *Acidobacteria* was depressed in obese individuals when compared with healthy controls [27]. The relative abundance of *Fusobacteria* showed a gradual upwards trend in Alzheimer’s patients [28]. Recently, it had been revealed that the relative abundance of *Fibrobacteres* was remarkably reduced in enrolled cases infected with severe acute respiratory syndrome coronavirus 2 (SARS-CoV-2) [29]. Although these bacteria at the phylum levels might play important roles in the pathophysiology of other diseases, their roles in GBS infection-related neonatal diseases have yet to be fully elucidated. Therefore, elucidating the relationship between these abnormal microorganisms and neonatal diseases caused by GBS infection may be a topic worthy of further research. 

Complete blood cell count (CBC) is one of the most frequently ordered laboratory tests. It is drawn for several times during pregnancy. CBC examination is routinely performed in early pregnancy, second trimester pregnancy and third trimester pregnancy to assess anemia, infection and other related conditions during pregnancy. CBC was related to early onset sepsis (EOS) of newborns during the first week of life [30]. Maternal inflammatory response is closely related to the fetal inflammatory response [31].In this study, although no difference in the CBC test was observed between GBS-positive and GBS-negative pregnant women, the level of OTU122 was positively correlated with monocyte count values, and the level of OTU518 was positively associated with basophil count values in pregnant women. Conversely, the level of OTU518 was negatively related to the proportion of monocytes, and the level of OTU375 showed a negative correlation with hemoglobin levels in GBS-infected pregnant women. These results suggest that the relative abundances of OTU122, OTU518 and OTU375 might be important players in the control of inflammatory state and hemoglobin levels. More clinical trials are needed to further confirm these findings. 

In 2021, Neonatal Resuscitation Group, Perinatal Medicine Branch, Chinese Medical Association released expert consensus on the clinical application of umbilical artery in newborn blood gas analysis (2021)[32]. Neonatal umbilical artery blood-gas analysis (umbilical arterial blood-gas analysis, UABGA), intrauterine and prenatal situation and Apgar score help to predicts the risk of the neonatal poor prognosis. Expert Consensus on the diagnosis of neonatal asphyxia in 2016 [33], noticed that the Apgar score alone had low specificity for diagnose neonatal asphyxia and UABGA had high specificity for it.

Additionally, we found that neonates born to mothers with GBS infection had significantly higher fractions of cord blood carboxyhemoglobin (COHb) and lower cord blood methemoglobin (MetHb) than those born to mothers without GBS infection. COHb, an indicator of endogenous carbon monoxide formation during the hem degradation process, could be used to confirm hemolysis in neonates [34]. MetHb is formed before the liberation of heme from hemoglobin, and then methemoglobin is increased. This indicates increased CO production and heme availability [35]. It is likely that both COHb and MetHb could serve as useful biomarkers of neonatal diseases caused by GBS colonization, and this hypothesis deserves further confirmation. Additionally, we found that cord blood levels of COHb, MetHb, PCO_2_, PH and ABE had significant correlations with the relative abundances of OTU80, OTU518, OTU122 and OTU375. These results suggest that the intestinal microbiota disturbance caused by GBS infection may affect biochemical indicators of neonatal umbilical cord blood. Whether GBS colonization-induced gut microbiota dysbiosis in pregnant women would be involved in neonatal circulatory diseases and, therefore, various infections is unknown, and more evidence is still needed.

Overall, our present study showed the changes in the gut microbiota of pregnant women following GBS infection, and provided novel evidence in support of intestinal microbiota dysbiosis in mothers intestinal systems after GBS colonization. Moreover, gut microbiota dysbiosis may be closely associated with several neonatal umbilical artery blood biochemical indicators in a Chinese cohort, which paved the way for a larger cohort-based clinical validation study and raised the recognition of the critical role of maternal gut microbiota imbalances in neonatal blood biochemical markers and infectious diseases. Of note, it is warranted to examine whether application of therapeutic probiotics could inhibit the colonization of GBS in pregnant women with fewer or without side effects, thereby reducing the occurrence and development of neonatal infectious diseases caused by GBS infection. 

## Figures and Tables

**Figure 1 pathogens-11-01297-f001:**
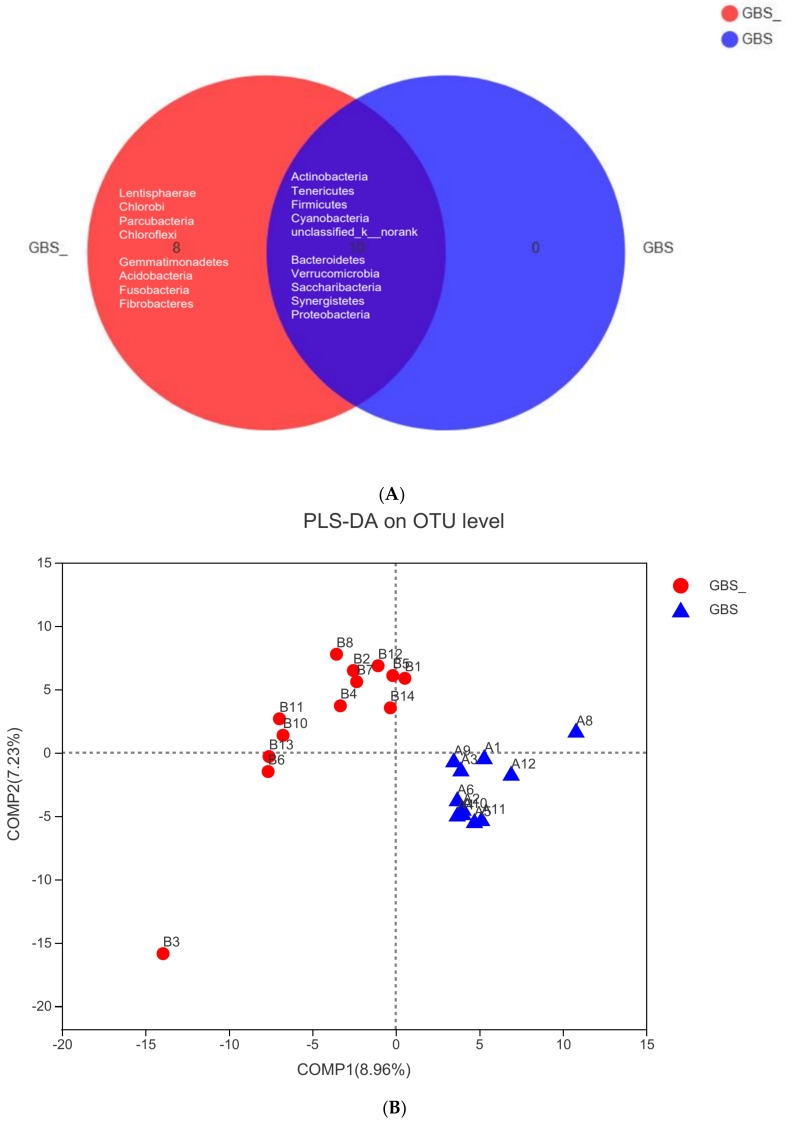
16S rDNA analysis of gut microbiota. (**A**) Venn map analysis of species; (**B**) OUT-based PLS-DA analysis; (**C**) community bar plot on genus level; (**D**) analysis of species differences between the GBS- and GBS+ groups at genus level. OTU, operational taxonomic unit; PLS-DA, partial least squares-discriminate analysis; GBS, Group B *Streptococcus*. The red region is the GBS-negative group (GBS-) and the blue is the GBS-positive group (GBS+).

**Figure 2 pathogens-11-01297-f002:**
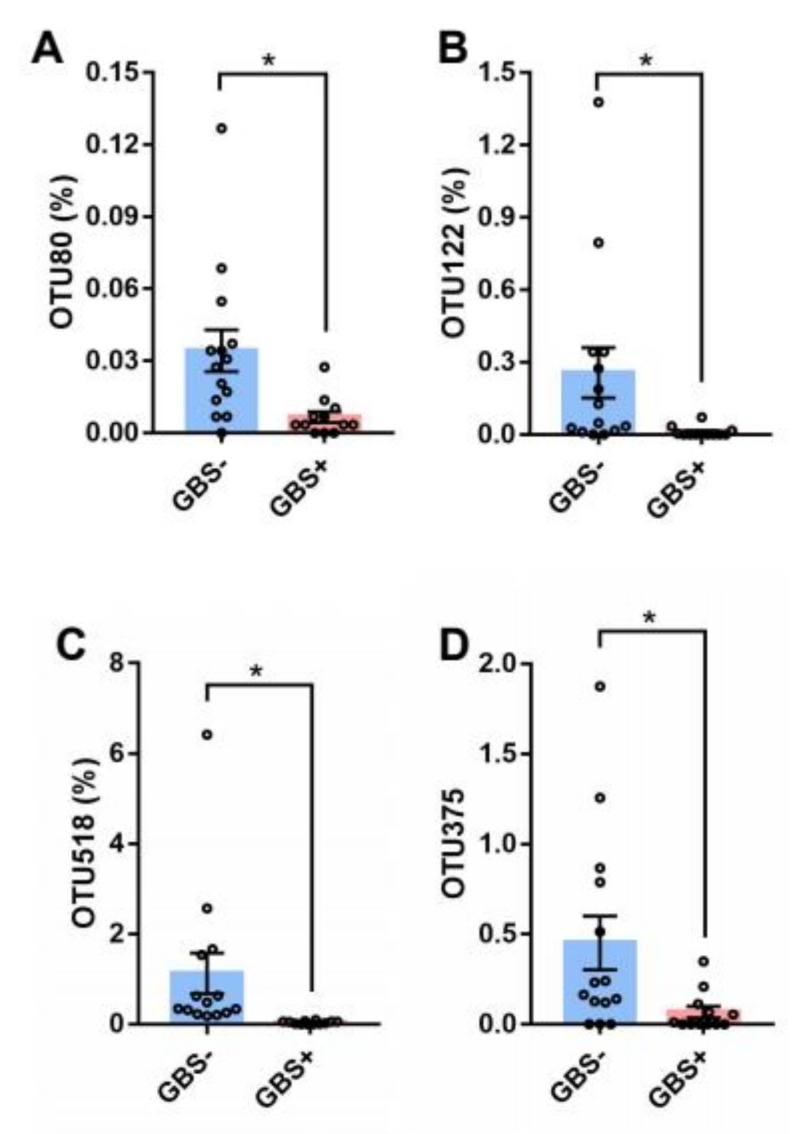
OTU richness of GBS− and GBS+ groups. (**A**–**D**) Bacterial species from GBS+ groups had the greatest decrease in abundance compared with GBS- groups. All values represent means ± SEMs. * *p* < 0.05 by Student’s t test. OTU, operational taxonomic unit; GBS, Group B *Streptococcus*.

**Figure 3 pathogens-11-01297-f003:**
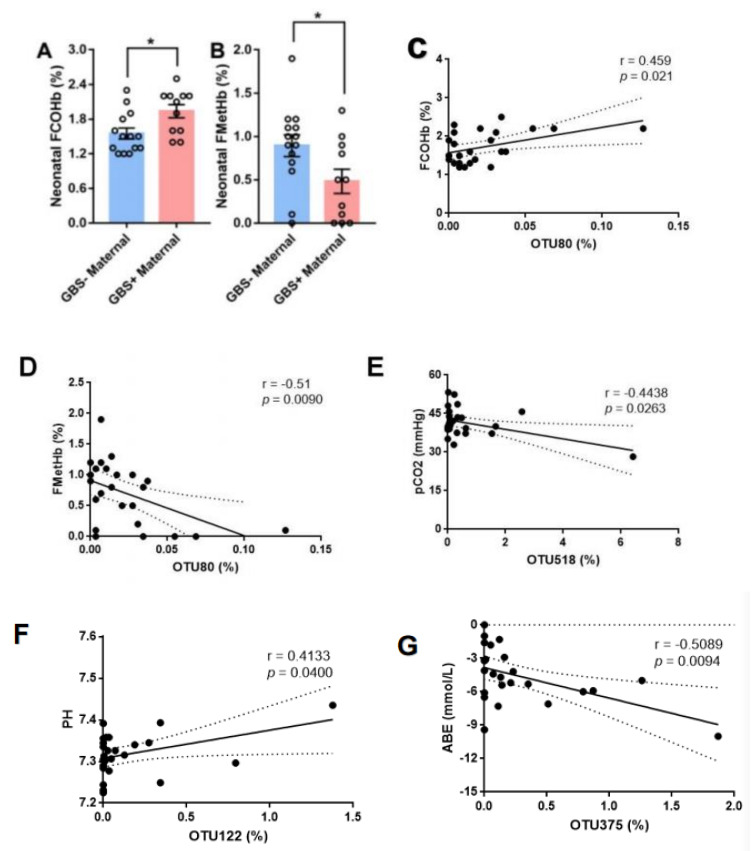
Neonatal umbilical arterial blood-gas analysis. (**A**) The levels of neonatal blood-gas FCOHb. All values represent means ± SEMs. * *p* < 0.05 by Student’s t test; (**B**) The levels of neonatal blood-gas FMetHb. All values represent means ± SEMs. * *p* < 0.05 by Student’s t test. (**C**) FCOHb (y-axis) and OTU80 (x-axis); significant positive correlation, r = 0.459, *p* = 0.021. (**D**) FMetHb (y-axis) and OTU80 (x-axis); significant negative correlation, r = −0.51, *p* = 0.009. (**E**) pCO_2_ (y-axis) and OTU518 (x-axis); significant negative correlation, r = −0.4438, *p* = 0.0263. (**F**) PH (y-axis) and OTU122 (x-axis); significant positive correlation, r = 0.4133, *p* = 0.04. (**G**) ABE (y-axis) and OTU80 (x-axis); significant negative correlation, r = −0.5089, *p* = 0.0094. FCOHb, fractions of carboxyhemoglobin; FMetHb, fractions of methemoglobin; ABE, actual base excess; OTU, operational taxonomic unit; GBS, Group B *streptococcus*.

**Figure 4 pathogens-11-01297-f004:**
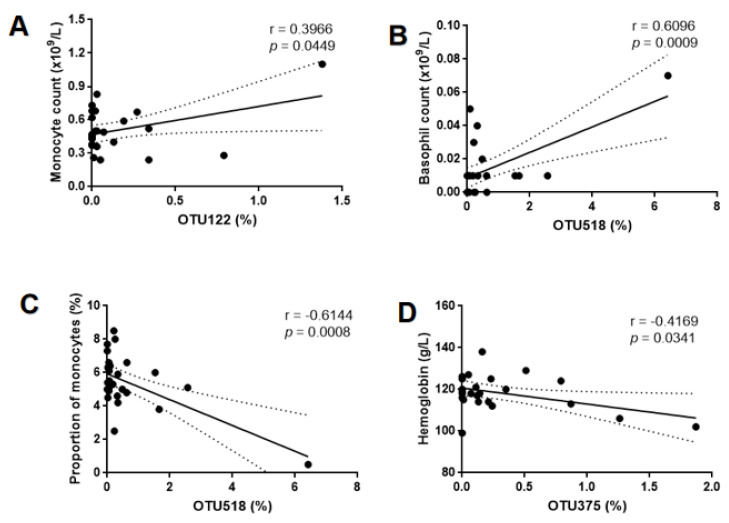
Correlation analysis between maternal inflammatory indexes and OUT level. (**A**) Monocyte count values (y-axis) and OTU122 (x-axis); significant positive correlation, r = 0.3966, *p* = 0.0449. (**B**) Basophil count value (y-axis) and OTU518 (x-axis); significant positive correlation, r = 0.6096, *p* = 0.0009. (**C**) Proportion of monocytes (y-axis) and OTU518 (x-axis); significant negative correlation, r = −0.6144, *p* = 0.0008. (**D**) Hemoglobin (y-axis) and OTU80 (x-axis); significant negative correlation, r = −0.4169, *p* = 0.0341. OTU, operational taxonomic unit.

**Table 1 pathogens-11-01297-t001:** Comparison of clinical, demographic and blood routine data of GBS^−^ and GBS^+^ subgroups.

Characteristic	GBS− (n = 14)	GBS+ (n = 12)	*p*-Value
Age (years)	29.79 ± 2.97	30.58 ± 3.96	0.563
Gender	Female	Female	-
WBC (×10^9^/L)	8.53 ± 1.20	9.37 ± 2.76	0.309
HGB (g/L)	118.43 ± 8.21	118.67 ± 8.90	0.944
Neutrophil (%)	72.21 ± 5.20	71.64 ± 9.48	0.849
Basophil (%)	0.20 ± 0.30	0.17 ± 0.16	0.819
Lymphocyte (%)	20.75 ± 4.87	21.46 ± 8.41	0. 792
Monocyte (%)	5.75 ±0.10	5.09 ± 0.22	0.327
Eosinophil (%)	0.97 ± 0.46	1.18 ± 1.11	0.536
Neutrophil count (×10^9^/L)	6.19 ± 1.17	6.80 ± 2.29	0.393
Basophil count (×10^9^/L)	0.01 ± 0.01	0.02 ± 0.02	0.704
Lymphocyte count (×10^9^/L)	1.74 ± 0.32	1.92 ± 0.67	0.370
Monocyte count (×10^9^/L)	0.49 ± 0.11	0.52 ± 0.27	0.668
Eosinophil count (×10^9^/L)	0.10 ± 0.04	0.10 ± 0.07	0.944

Data are presented as mean ± SD and were analyzed by Student’s t-test. WBC, white blood cell; HGB, hemoglobin.

## Data Availability

Data are contained within the article.

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
