# Peer review of "Association of Gut Microbiota Composition in Pregnant Women Colonized with Group B Streptococcus with Maternal Blood Routine and Neonatal Blood-Gas Analysis"

_pathogens, 2022, doi:10.3390/pathogens11111297_

Round 1
Reviewer 1 Report
In manuscript "Association of gut microbiota composition in pregnant women colonized with group B streptococcus with maternal blood routine and neonatal blood gas analysis" authors for the first time describes that the altered gut microbiota compositions were related with the inflammatory state in GBS-positive pregnant women and neonatal blood gas indicators. The paper is correctly written and the results are clear and the correct conclusions are interpreted from them.
I have a few small remarks.
Instead of the term flora, the term microbiota should be used.
Check the new nomenclature for the genus Lactobacillus.
The figures should be checked because they are not sharpened.
For Figure 1, in the description below the picture, explain what the pictures show and increase the font in the graphs because it is too small in some places.
Check references (font and writing style).
Reviewer 2 Report
Abstract: Incomplete. Missing study design and clarity about measures. It would be preferable to follow a structure for the abstract.
Introduction:
*Please describe if this study done in a setting where universal antenatal GBS screening is performed? What is the prevalence of GBS is the study setting?
*In 2019, the ACOG revised the guideline for Universal Screening for Antenatal GBS to 36 weeks. Please indicate the years of the study so that the ready can see that data collection occurred before that important change.
*Your exploration of the CBC with diff as an outcome is intriguing. Most of the scientific literature on GBS and the perinatal microbiome does not use the CBC with diff as an "inflammatory marker." Rather other indices (such as Cytokines) are used. Given that, please provide a background literature that supports your use of the CBC with diff, besides the fact that it is routine. When was the CBC drawn during pregnancy? Throughout the article you refer to these by different names "inflammatory indexes" "inflammatory indicators."
*Similarly, Umbilical cord gases are not performed routinely in most parts of the world and generally not used as an outcome for perinatal microbiome studies. The use of this measure as an outcome in your study is intriguing. Please provide some literature that led you to explore umbilical cord gases as an outcome.
Method:
*Please clearly describe the study sample and methods used. Was this a convenience sample? A convenience sample of GBS positive participants matched to a cohort of controls? Need more detail of the sample and design to understand the results. How were participants consented and enrolled? This is a small descriptive study and when presenting and discussing findings that context should be clear.
Results:
The figures all contain unclear markers for GBS positive vs GBS negative.
*Line 64 "Coincidentally, the gut microbi-ome has been recognized as a potential novel player in potential prenatal and early life of infant" is unclear and an understatement.
Lines 113-114 "All pregnant women are routinely tested for blood routine during 37 to 38 weeks, reflecting the normal physiological changes during pregnancy. We don’t limit their fast-114 ing state, in case of hypoglycemia in pregnant women." Unclear first sentence. "Blood routine" is an unclear phrase. If a test reflects normal physiologic changes of pregnancy, then why do it? This reviewer assumes it is done to identify anemia before the labor and birth. Switched to first person. The line about limiting fasting is unclear because there is no need to fast for a CBC with diff. Unless what the authors mean is that the CBC is done routinely at 26-28 week gestation.
Discussion
*No need to refer to "the present study."
*The first paragraph of the discussion is unclear due to word choice problems and run on sentences. This is the first use of "Chinese cohort."
*Overall the discuss would be more readable with better organization and use of more concise sentences. Generally begin with major findings and next compare the existing literature.
